# Comparison of two popular transducers to measure sit-to-stand power in older adults

Anoop T. Balachandran[1]*, Samuel T. Orange[2], Yipeng Wang[3], Renee Lustin[1], Andy Vega[1], Norberto Quiles[1]

1 Department of Family, Nutrition and Exercise Sciences, Queens College, New York, NY, United States of America, 2 Department of Sport, Exercise and Rehabilitation, Faculty of Health and Life Sciences, Northumbria University, Newcastle Upon Tyne, United Kingdom, 3 Department of Biostatistics, University of Florida, Gainesville, FL, United States of America

* athozhuthungalba@qc.cuny.edu

**Data Availability Statement:** https://osf.io/p78mb 10.17605/OSF.IO/Q8PSA.

**Funding:** AB received funding from Professional Staff Congress (PSC) at the City University of New York (Award Number: 64115-0052). https://www.

## Abstract

The Tendo Unit (TU) and GymAware (GA) are the two most frequently used linear transducers for assessing muscle power in older adults via the sit-to-stand (STS) test. Unlike TU, GA incorporates a sensor mechanism to correct for non-vertical movements, which may lead to systematic differences between devices. The aim of this study therefore was to compare GA to TU for measuring STS power in community-dwelling older adults. Community-dwelling adults (n = 51, aged ≥65 years, 61% female) completed a single chair stand, with peak power measured simultaneously using GA and TU. Participants also completed the pneumatic leg press, 8-Foot Up and Go (TUG) test, Short Physical Performance Battery (SPPB), and self-reported measures of physical function. Intraclass correlations (ICC) were used to assess agreement, and Pearson's correlations were used to assess correlations. The study protocol was prospectively registered on the Open Science Framework. In alignment with our pre-registered hypothesis, peak power demonstrated an ICC of 0.93 (95% CI: 0.88, 0.96). For secondary aims, both transducers showed a correlation greater than 0.8 compared to pneumatic leg press power. For physical performance outcomes, both TU and GA showed similar correlations, as hypothesized: SPPB (r = 0.29 for TU vs. 0.33 for GA), Chair Stands (r = −0.41 vs. -0.38), TUG Fast (r = −0.53 vs. -0.52), mobility questionnaire (r = 0.52 vs. 0.52) and physical function questionnaire (r = 0.44 vs. 0.43). GA and TU peak power showed a high degree of agreement and similar correlations with physical and self-reported performance measures, suggesting that both methods can be used for assessing STS power in older adults.

## Introduction

Over the last two decades, muscle power—defined as the rate of mechanical work—has emerged as a key factor influencing physical function in the aging population: As individuals age, power experiences a more rapid decline than strength [1, 2], and demonstrates a stronger correlation with both physical function [3, 4] and risk of falls compared to strength alone [5].

rfcuny.org/RFWebsite/ The funders had no role in study design, data collection and analysis, decision to publish, or preparation of the manuscript.

**Competing interests:** The authors have declared that no competing interests exist.

Notably, numerous systematic reviews [6, 7] and meta-analyses [8, 9] now confirm that power training improves physical function to a greater extent in older adults than traditional strength training.

Measuring muscle power during a single chair stand, known as the sit-to-stand (STS) power test, offers a low-cost, accessible, valid and portable method for evaluating lower body muscle power [10–12]. This assessment quantifies power in a functional context, specifically tailored to upright, weight-bearing, everyday activities such as walking or stair climbing. The majority of tools designed for measuring lower body power are expensive and demand significant space and expertise to operate, such as Nottingham Power Rig [13], force plates [14], 3D motion capture systems [15], and isokinetic dynamometry measurements [16]. Considering the importance of power in older adults, coupled with the ease of use, low cost, and portability, the STS test is a highly promising method for assessing power and predicting disability, falls, and other adverse outcomes in older adults.

The Tendo Unit (TU) and GymAware (GA) are the two most frequently used linear transducers in research [17, 18] and practical settings [19]. Although both are linear position transducers, there are technical differences that could impact the power measurements. Unlike other LPTs, GA utilizes a unique optical sensor to measure the angle of lift and account for any non-vertical movement. Consequently, the power recorded automatically corrects for any horizontal displacement and is solely derived from the vertical component. This difference can have a significant impact on STS power because older adults often use a strategy known as "momentum transfer" when rising from a chair, which involves a horizontal momentum shift from the trunk to standing [20]. However, TU does not incorporate any adjustments and includes the horizontal displacement for power calculations. This major discrepancy between devices, in addition to potential differences in the acquisition of displacement data and double differentiation procedures, may lead to systematic differences between GA and TU. Systematic differences between measurement tools would make it extremely challenging to use the STS power test to compare power across studies and identify cut points for screening adults at high risk of disability, falls and other adverse outcomes related to low muscle power. Furthermore, differences in power can also significantly affect the association between power and functional outcomes, thereby impacting the predictive validity of these devices.

Despite the widespread use and inherent differences, no study has compared these two linear transducers in the context of measuring STS power in older adults. Therefore, the main aim of this study was to compare the GA to the TU for measuring STS power in community-dwelling older adults. We also examined the convergent validity of GA and TU by assessing the relationship between STS power recorded by both devices with the pneumatic leg press, and examined whether STS power was related to measures of physical function.

As stated in the pre-registration plan (https://osf.io/3sh5g), our hypotheses were:

- Primary: STS peak power and average power recorded by the GA and TU will demonstrate high agreement (intraclass correlation coefficient (ICC) ≥ 0.75).

- There will be a positive Pearson's correlation (ICC ≥ 0.8) between STS peak power recorded by the linear transducers and pneumatic leg press power (convergent validity)

- STS peak power recorded by GA and TU will demonstrate comparable correlations, differing by no more than 0.1, with measures of physical function, encompassing both performance-based assessments and patient-reported outcomes (convergent validity).

## Materials and methods

### Participants

Participants were enrolled from the local New York community using flyers, posters, and advertisements in newsletters from September 8, 2022, till June 1, 2023,. The inclusion criteria required volunteers to be aged 65 years or older, live independently in the community, and be able to communicate in English. Exclusion criteria included severe knee arthritis (either osteo-arthritis or rheumatoid arthritis) that could be exacerbated by exercise, and serious neurological disorders such as Parkinson's disease. Temporary exclusion criteria encompassed major surgery or fracture of the hip or knee, hip/knee replacement, or hospitalization in the last 6 months, as well as a history of heart attack or heart disease, major heart surgery, valvular disease, or stroke in the past 6 months. The protocol received approval from the University's Institutional Review Board, and all participants provided written and signed informed consent before participating.

Test instructions and procedures were standardized, and the research staff underwent training prior to the study. Participants made one visit to the laboratory, during which they provided informed consent, reported baseline characteristics, had their weight and height measured, and were tested on the following measures: All the following tests for all the participants were conducted in the order given below:

### Pneumatic leg press

The pneumatic LP is considered valid, reliable, and is the most commonly employed method to measure lower body power [17, 21]. This pneumatic equipment utilizes pressurized air cylinders to provide resistance, a departure from traditional machines that use weight plates. Following a demonstration of the proper technique by the tester, participants engaged in 3–5 warm-up repetitions with 50% of their body weight and 1–2 repetitions with their full body weight using a pneumatic leg press (Keiser A300, Keiser Sports Health Equipment, Fresno, CA). The machine was adjusted to ensure a sitting knee angle of 90 degrees flexion. In instances where participants reported pain or were unable to maintain the position due to anatomical restrictions, the seat was adjusted to the next closest setting to 90 degrees. After the warm-up, resistance was set to their body weight, and participants were instructed, "When you are ready, push as fast as possible," emphasizing a slow, controlled movement during the lowering phase. The software calculated work and power during the concentric phase of each repetition by sampling the system pressure (from which force is calculated) and position at 400 Hz. The highest peak power and average power across three repetitions, with 1 minute of rest between sets, were used for the final analysis. Peak power was chosen as the primary outcome since a majority of the studies using the pneumatic leg press have used peak power as the power outcome [22, 23] and studies using the STS power test showed similar correlations for peak power and mean power when compared to functional outcomes [12].

### Gym aware and Tendo Unit

The STS power test consisted of participants standing up from the chair once while attached to the linear transducers, in line with previous protocols [10, 19]. After the tester demonstrated the proper technique, participants performed 3–5 warm-up sit-to-stands at a normal speed before initiating the power tests. A belt was securely fastened around the participant's waist, above the Iliac crest. Strings from the two units (Tendo Weightlifting Analyzer, Trencin, Slovak Republic), and Gym Aware LE (Kinetic Performance Technology, Canberra, ACT, Australia) were attached to the belt, ensuring that the strings were as close to perpendicular to the

floor as possible when the participant stood up. The chair height was set at 45 cm. Participants were instructed to sit in the middle or on the edge of the chair to minimize forward trunk lean. Participants began in a seated position with their arms folded across their chest, stood up as quickly and safely as possible, and then returned to the seated position. The standard instruction given before each sit-to-stand was, "When you are ready, get up as fast as you can." For both units, power was calculated by the software and displayed based on the vertical velocity (m/s) and the mass moved (kg) during the standing portion of the test. GA uses variable rate sampling and further down-samples to a maximum of 50 points per second without filtering. TU also uses a variable sampling rate with an adjustable filter to remove unwanted movements. Instantaneous velocity was estimated as change in displacement over time, and acceleration was calculated as the change in velocity over the change in time. Force was estimated by the product of body mass and acceleration, and power was calculated as the product of force and velocity. GA was tethered to an iPad via Bluetooth, running the GA app Version 4.1.5, while the TU was connected to a desktop computer via Bluetooth, running the TU software Version PA 7.1.3. The highest peak power and the average across three sit-to-stands, with 1 minute of rest between stands, were used for analysis.

## SPPB

The Short Physical Performance Battery (SPPB) is extensively utilized in multi-center trials to assess physical function in older adults [24, 25]. The SPPB has demonstrated reliability and validity in predicting institutionalization, mortality, and disability [26, 27]. This battery comprises three tests:

a. Walk speed: A 4-meter walk conducted at the usual pace, with the faster time out of two trials recorded.

b. Balance: Three standing balance tests (narrow stance, semi-tandem, and tandem), each lasting 10 seconds. The total time per test was recorded.

c. Chair stands: One trial consisting of five chair stand tests performed as quickly as possible, with the total time recorded.

Based on the completion time, each of the three tests is scored between 0 and 4, and these scores are then summed to achieve a maximum score of 12 for the total SPPB score. Higher scores indicate better physical performance. The walk speed and chair stand outcomes used for analysis were derived from the SPPB.

## Timed up and go (TUG)

The timed up and go (TUG) measures dynamic balance and agility in older adults [28]. The test involves standing from a chair, walking around a cone 3 m away, and sitting back down. We performed the test at both usual pace and fast pace. There were two trials per pace with 1 min rest, and the faster time was recorded.

## Patient-Reported Outcome Measures (PROMs)

Unlike performance measures, self-reported measures evaluate an individual's functional capacity in their actual or lived environment, a methodology increasingly acknowledged by governmental regulatory agencies for a comprehensive assessment of function [29]. We employed the Patient Reported Outcomes Measurement Information System (PROMIS) physical function and mobility questionnaire developed by the National Institutes of Health (NIH). PROMIS utilizes item response theory and computerized adaptive testing to maximize efficiency and has shown to be reliable and valid in a large sample of the general population [30, 31]. PROMIS measures use a T-score metric in which 50 is the mean of a relevant reference

population and 10 is the standard deviation (SD) of that population. High scores mean more of the concept being measured (e.g., more mobility, more physical function). Participants completed the questionnaires using the PROMIS iPad App without assistance from the study staff.

### Sample size

We used an estimation-based approach to calculate sample size for our primary outcome: Based on a 95% confidence interval (CI) of 0.75–0.95 ICC (i.e. 0.75 is the lower bound of the interval) and an 80% probability of the desired precision of 0.2 (total width) with two repeats, we calculated the required sample size to be 45 participants [32]. Assuming a 10% drop due to missing values, we would need around 49 participants.

### Statistical analysis

Continuous variables were expressed as mean (SD) and categorical variables were presented as frequencies and percentages. Data were imported into R (version 4.0.0) for analysis. In accordance with the pre-registration, for the primary hypothesis, ICCs with 95% CIs were calculated using a two-way random-effects model of absolute agreement single measure for peak and average power (ICC (2,1)) [33]. CIs were constructed using the bias-corrected and accelerated bootstrap with 10,000 replicates due to non-normal data. For interpreting ICC, we used <0.5 —poor, 0.5–0.75—moderate, 0.75–0.9—good, >0.90—excellent [34]. For other outcomes, Pearson's correlation coefficient (r) with 95% CIs was used to examine correlations between STS power and LP power, and physical performance and self-reported measures. A Bland-Altman (BA) plot was used to evaluate the mean difference and limits of agreement (LOA) [35].

## Results

The demographics and characteristics of the 51 participants are shown in Table 1. The mean age of the participants was 70.8 ± 8.4 years, 61% females with a mean Body Mass Index (BMI) of 25.5 ± 4.6 kg/m$^2$. The participants were high physically functioning as shown by mean score of 11.3 ± 1.1 on the SPPB. Previous research has classified an SPPB score greater than 10 as high functioning [36]. The box plot for power values for TU and GA is shown in Fig 1.

### Agreement between devices

For our primary aim of assessing agreement between devices, peak power showed an excellent ICC of 0.93 with 95% CI ranging from 0.88 to 0.96 (Table 2). ICC for average power was 0.81 with 95% CI ranging from 0.73 to 0.87. Hence, both peak and average power met our pre-defined criteria for acceptable agreement of ICC ≥ 0.75.

### Comparison between devices and physical performance measures

For physical performance outcomes and self-reported outcomes, both TU and GA showed similar correlations (within 0.1) for peak power as hypothesized in Table 3: SPPB (r = 0.29 (TU) vs. 0.33 (GA)), chair stands (− 0.41 vs. -0.38), TUG normal (− 0.46 vs. -0.48), TUG fast (− 0.53 vs. -0.52) and balance (0.12 vs. 0.16), and for PROMs mobility questionnaire (0.52 vs. 0.50) and physical function questionnaire (0.44 vs. 0.43).

**Table 1. Participant characteristics.**

|  | N = 51 |
|---|---|
| Age, y | 70.8 (8.4) |
| Gender |  |
| Male | 20 (39.2%) |
| Female | 31 (60.8%) |
| Peak Power |  |
| TU peak power, W | 817.7 (338.6) |
| TU avg. power, W | 384.9 (116.6) |
| GA peak power, W | 924.1 (375.7) |
| GA avg. power, W | 440.3 (139.7) |
| BMI, kg/m$^2$ | 25.5 (4.6) |
| Physical Function |  |
| SPPB score, s | 11.3 (1.1) |
| Chair Stand, s | 9.6 (1.9) |
| TUG fast, s | 6.3 (1.2) |
| TUG slow, s | 8.0 (1.8) |
| PROMIS Mobility t-score | 50.6 (7.2) |
| PROMIS Function t-score | 52.2 (6.4) |
| Race/Ethnicity |  |
| White | 39/51 (76.5%) |
| African American/Black | 2/51 (3.9%) |
| Asian | 10/51 (19.6%) |
| Other | 0/51 (0%) |
| Income (<$75,000/year) | 15/51 (29.4%) |
| College education | 17/51 (33.3%) |
| Falls | 0.2 (0.4) |
| Conditions, No./total (%) |  |
| Hypertension | 19/51 (37.3%) |
| Heart Condition | 16/51 (31.4%) |
| Diabetes | 4/51 (7.8%) |

## Comparison between devices and LP power

As shown in Table 4, Pearson's correlations (r) for peak power between TU [0.83 (0.72, 0.90)] and GA [0.84 (0.73, 0.91)] and LP power showed correlations greater than 0.80. As shown in Fig 2, Pearson's correlations for peak power between TU and GA showed 0.96 (0.93, 0.98).

BA plots with LOA are reported in Figs 3 and 4.

## Missing data

We omitted two participants from the analysis due to technical issues with the GA.

## Discussion

Our study comparing TU and GA met all our pre-defined criteria for acceptable agreement and convergent validity. The units showed moderate to excellent agreement, were correlated with LP peak power, and had similar correlations with physical and self-reported performance measures. These results suggest that both methods can be used for assessing STS peak power in older adults. However, average power did not meet our pre-defined criteria for acceptable agreement.

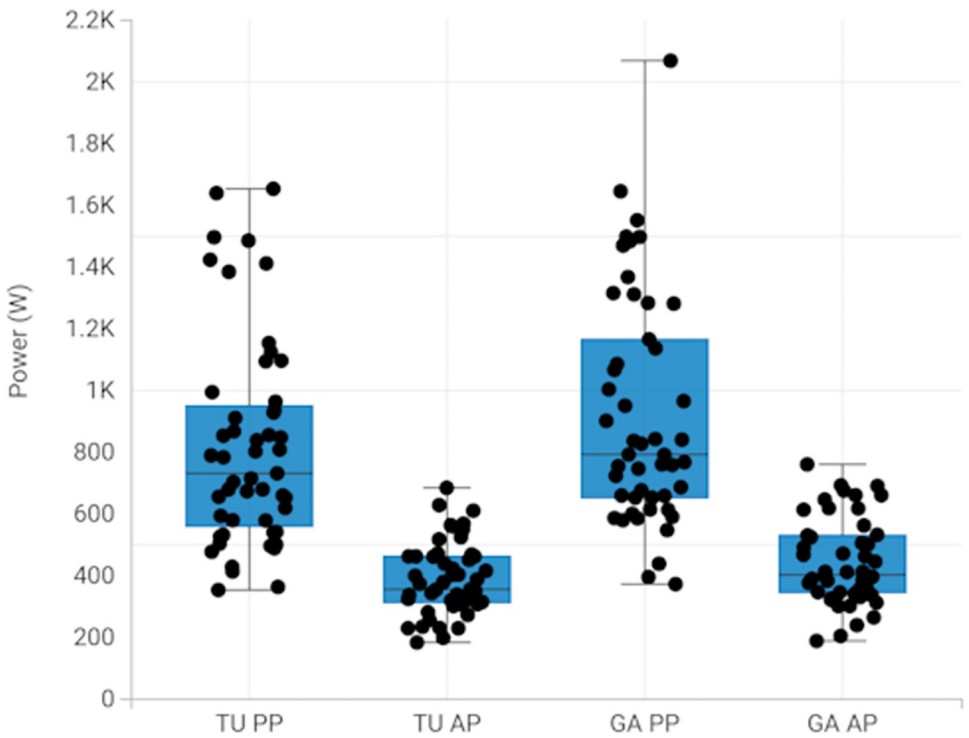

**Fig 1. Box plots for TU and GA peak power and average power.**

Several thresholds in the literature exist for interpreting agreement using ICC. We chose a commonly used threshold that shows ICC < 0.5 indicates poor reliability, between 0.5 and 0.75 indicates moderate reliability, between 0.75 and 0.9 indicates good reliability and > 0.90 indicates excellent reliability [34]. Most of the reliability studies in sports and exercise literature have interpreted ICC threshold based on the point estimate while ignoring the 95% CI or limits [32]. In the current study, we interpreted the ICC using both the point estimate and the interval bounds as recommended. For peak power, the point estimate of 0.92 shows excellent reliability, but the 95% CI (0.88, 0.96) shows that reliability could range anywhere from good to excellent. Average power showed a point estimate of 0.81 with 95% CI ranging from 0.73 to 0.87. The lower limit of the CI was below our threshold of 0.75 and thus the reliability for average power spanned from moderate to good.

To assess convergent validity, we compared the GA and TU peak power to LP peak power, which was used as our reference. As hypothesized, both GA (0.84, 95% CI 0.73, 0.91) and TU (0.83, 95% CI 0.72, 0.90) showed a correlation greater than 0.8. The correlation could largely be attributed to the differences in body positioning for the LP and the STS power test. The leg press is performed in a seated position with seat adjustments, while the chair stand test is done standing with no seat adjustment. We couldn't compare average power since our LP model only provides peak power values. In addition, all correlations compared to physical

**Table 2. Intraclass correlations (ICC) between GA and TU power.**

|  | ICC$_{\text{agreement}}$ |
| --- | --- |
| Peak Power | 0.93 (0.88, 0.96) |
| Average power | 0.81 (0.73, 0.87) |

**Table 3. Pearson's correlation coefficient of GA and TU compared to physical performance and self-reported outcomes.**

| | Physical Performance Outcomes | | | | | Patient-Reported Outcomes (PROMIS) | |
|---|---|---|---|---|---|---|---|
| | SPPB | Chair Stands | $TUG_{normal,}$ | $TUG_{fast,}$ | Balance | Mobility | Function |
| $GA_{pp}$ | 0.33 (0.06, 0.56) | -0.38 (-0.59, -0.11) | -0.48 (-0.67, -0.23) | -0.52 (-0.70, -0.28) | 0.16 (-0.12, 0.42) | 0.50 (0.25, 0.69) | 0.43 (0.17, 0.64) |
| $TU_{pp}$ | 0.29 (0.01, 0.52) | -0.41 (-0.62, -0.15) | -0.46 (-0.65, -0.21) | -0.53 (-0.70, -0.30) | 0.12 (-0.16, 0.39) | 0.52 (0.28, 0.70) | 0.44 (0.18, 0.64) |

**Table 4. Pearson's correlation coefficient between LP and TU and GA for peak power.**

| | $LP_{PP}$ | $GA_{PP}$ |
|---|---|---|
| $GA_{PP}$ | 0.84 (0.73, 0.91) | – |
| $TU_{PP}$ | 0.83 (0.72, 0.90) | 0.96 (0.93, 0.98) |

performances and self-reported measures were higher for both GA and TU than for LP power. We believe that the higher correlations could partly be explained by the ground-based or functional nature of the sit to stand test compared to the seated LP movement [37–39].

The GA unit showed higher values for both peak and average power compared to TU. However, the difference in power did not have an impact on the correlations to performance and self-reported measures. It should be noted that a higher or lower power does not necessarily indicate that one unit is more accurate or true than the other. This would depend on other psychometric measures such as the convergent validity, reliability, and responsiveness of the unit. All correlations compared to performance and self-report measures ranged between 0.3

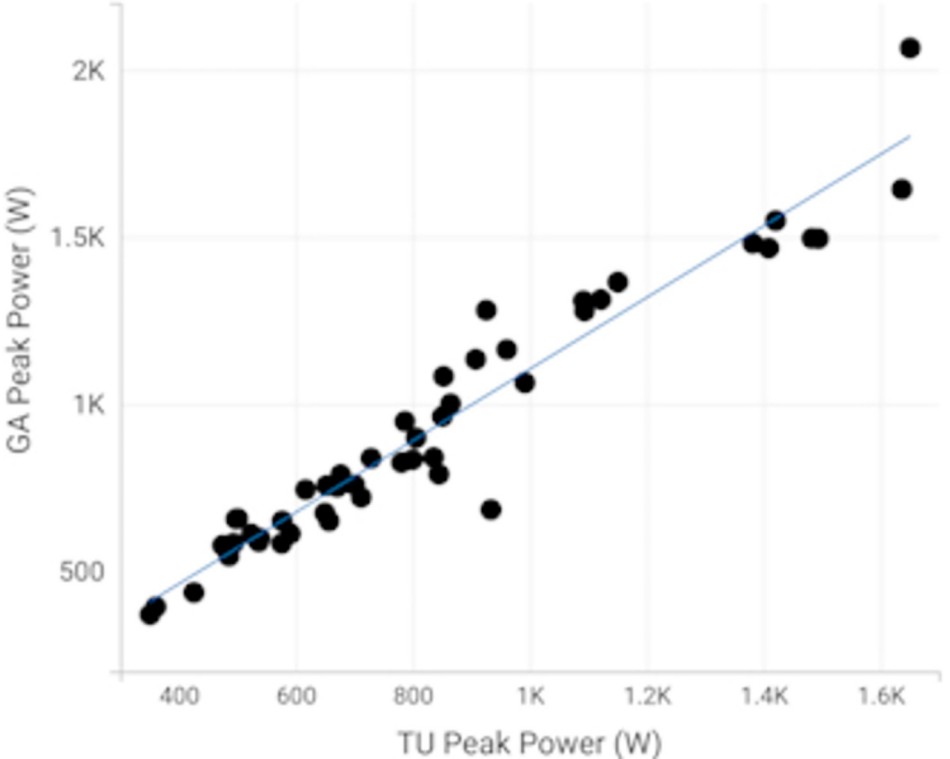

**Fig 2. Pearson's correlation coefficient between TU and GA for peak power.**

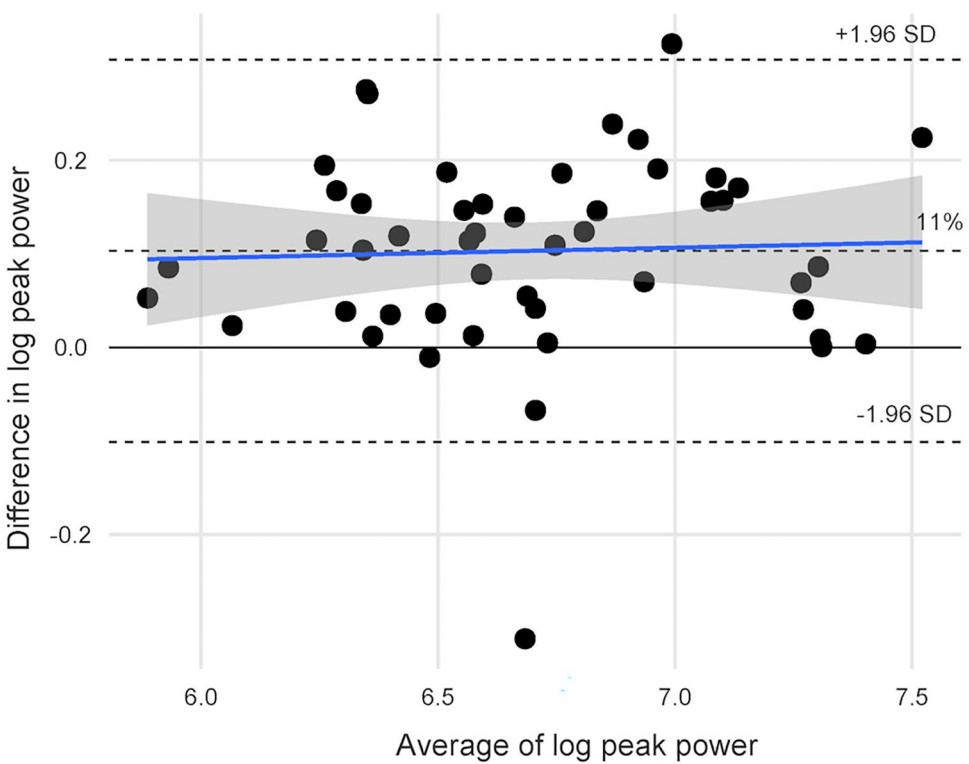

**Fig 3. Bland Altman (BA) plot for peak power after logarithmic transformation.**

to 0.5 and differences between GA and TU were well within the range of 0.1 as hypothesized. These correlations are similar to other studies comparing linear transducers to physical performance measures [10, 12, 40]. For example, another study comparing TU showed a correlation ranging from 0.3 to 0.4 compared to performance and self-reported measures [10]. Small differences in correlations between studies could be primarily explained by a combination of factors, including functional status, age, and the methods used in each study. Considering our results and those from other studies that investigated the validity of TU and GA independently, the convergent or construct validity of both devices is adequate and on par.

The BA plot shown in Figs 3 and 4 was used to assess the measurement error or absolute reliability. The BA plots showed heteroscedasticity since the scatter of differences increases with higher values of power. As recommended [35], we applied a logarithmic transformation and back transformed the log values to obtain a mean difference of 1.11 and limits of agreement of 0.90 and 1.36. In other words, the GA peak power is 1.11 times (11%) higher than TU power with limits of agreement 0.90 and 1.36 (10% lower to 36% higher) times than TU power. All data points, except for three, consistently showed a higher value for GA than TU. For average power, back transformed log values showed a mean difference of 1.12 (12%) and limits of agreement of 0.860 and 1.47, which shows a mean difference of 12% with LOA as 14% lower and 47% higher. Compared to LP peak power, TU showed a mean difference of -80W, while GA showed a mean difference of 17W. If transducers are used sequentially or power comparisons are made across studies, power can be adjusted by adding or subtracting 11–12%. For instance, if the TU peak power is 500W, the GA peak power would be, on average, 11% higher, which translates to an additional 55W. Therefore, the estimated peak power of the GA transducer would be 555W. We did not set a threshold for measurement error in our

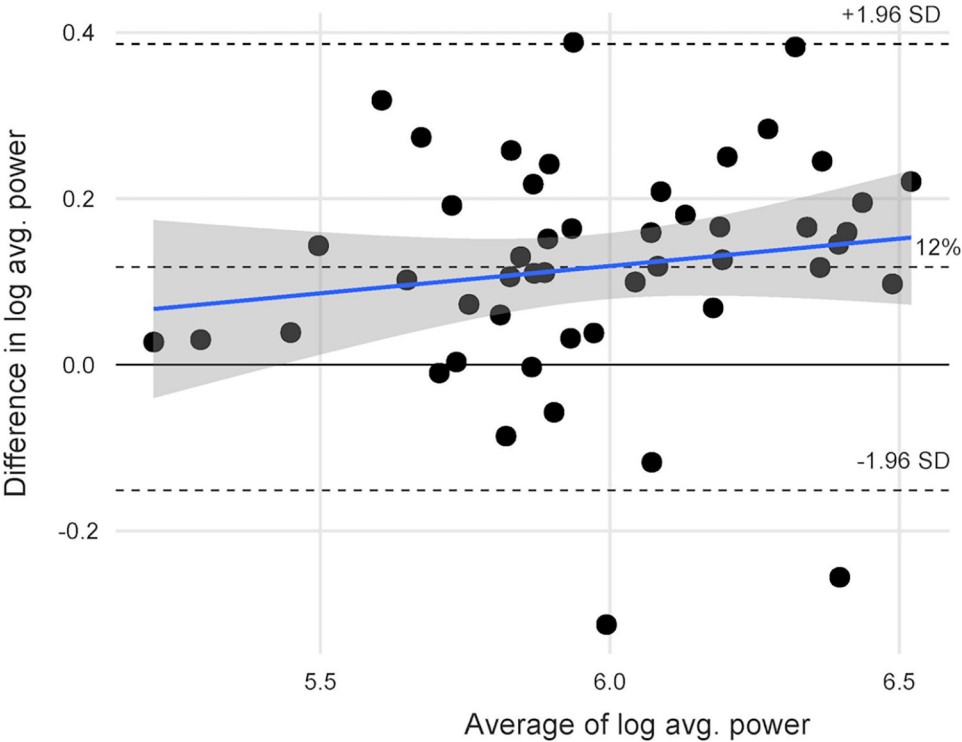

**Fig 4. Bland Altman (BA) plot for average power after logarithmic transformation.**

preregistration since minimal clinically important difference (MCID) or meaningful change of STS power test in community living older adults is unknown.

### Strengths and limitations

Our study was rigorously conducted. The major strength lies in the pre-registration of outcomes, hypotheses, and sample size estimates prior to data collection. Preregistration attenuates P-hacking and selective reporting [41]. Secondly, we adopted an estimation approach for the primary outcome of agreement that considers the lower and upper bound, rather than the point estimate, which is more commonly used in sports and exercise studies [25]. Our study also had several limitations. The MCID or meaningful change in power for the STS power test is not known. Without this knowledge, interpreting the differences in power measurements from both devices remains uncertain [42]. Additionally, we chose to use both transducers simultaneously, rather than on separate occasions, to minimize variations in effort and technique during the chair stand. However, accommodating two transducers might have slightly compromised the positioning of the devices and the string attachment to the waist, potentially impacting the power values. Based on the average SPPB score of 11.3, the majority of our participants could be classified as high-functioning with an average age of 70 years [36], meaning these results may not be extrapolatable to low-functioning older adults or those outside the average age group. We employed an estimation-based approach (i.e., considering the interval limits) for the primary outcome of the agreement. However, for our secondary outcomes, we relied on the point estimate for inferences, as the sample size was calculated based on the primary outcome. Lastly, we did not assess the test-retest reliability nor the responsiveness of the units which are important properties for a measurement device. Test-retest reliability for STS power test using the TU was reported previously as ICC of 0.96 (0.93, 0.97) [10], but test-retest

reliability for GA is unknown, although GA has shown high reliability in measuring power during resistance training movements [43].

In summary, GA and TU peak power showed a high degree of agreement and similar correlations with physical and self-reported measures of function. Practitioners and researchers can use both devices for measuring STS power in older adults, but should be cognizant of the absolute systematic differences when developing cut-off points to identify low muscle power or comparing power values in the literature.

## Author Contributions

**Conceptualization:** Anoop T. Balachandran, Samuel T. Orange, Norberto Quiles.

**Data curation:** Yipeng Wang.

**Formal analysis:** Yipeng Wang.

**Investigation:** Renee Lustin, Andy Vega.

**Methodology:** Anoop T. Balachandran, Samuel T. Orange, Norberto Quiles.

**Project administration:** Renee Lustin, Andy Vega.

**Supervision:** Anoop T. Balachandran.

**Visualization:** Yipeng Wang.

**Writing – original draft:** Anoop T. Balachandran.

**Writing – review & editing:** Anoop T. Balachandran, Samuel T. Orange, Yipeng Wang, Renee Lustin, Andy Vega, Norberto Quiles.

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
