## [Decision Letter · Decision Letter 0]

2 Jul 2024

PONE-D-24-08546Comparison of two popular transducers to measure sit-so-stand power in older adultsPLOS ONE

Dear Dr. Balachandran,

Thank you for submitting your manuscript to PLOS ONE. After careful consideration, we feel that it has merit but does not fully meet PLOS ONE’s publication criteria as it currently stands. Therefore, we invite you to submit a revised version of the manuscript that addresses the points raised during the review process.

We look forward to receiving your revised manuscript.

Kind regards,

Giuseppe Sergi

Academic Editor

PLOS ONE

Reviewers' comments:

Reviewer's Responses to Questions

**Comments to the Author**

1. Is the manuscript technically sound, and do the data support the conclusions?

Reviewer #1: Yes

Reviewer #2: Yes

2. Has the statistical analysis been performed appropriately and rigorously? 

Reviewer #1: Yes

Reviewer #2: Yes

3. Have the authors made all data underlying the findings in their manuscript fully available?

Reviewer #1: Yes

Reviewer #2: Yes

4. Is the manuscript presented in an intelligible fashion and written in standard English?

Reviewer #1: Yes

Reviewer #2: Yes

5. Review Comments to the Author

Reviewer #1: This paper compares for the first time two common methods to assess muscle power in older adults, the Tendo Unit (TU) and the Gym Aware (GA). As a second aim, it confirms the correlation between the STS peak power and physical performance measures AND LP power.

The findings are positive, with high degree of agreement ans similar correlations with physical function.

The study is based on a solid statistic method that includes a sample size estimate prior to data collection. Such occurence makes it hard to point out that my only comment is that I would have loved to see the results of such a precise and jolly design and method on a larger sample with age statification and frailty stratification. Other than that, I don't have major nor minor comments to submit.

(watch out for the typing error in the title)

Reviewer #2: The study by Balachandran et al. compared two popular assessment tools for skeletal muscle power in geriatric patients. The study is carefully done, and the results are sound and well-presented. All conclusions are supported by the data. There are certainly some limitations to this study that are adequately detailed in the discussion section. However, the manuscript needs minor improvements:

Page 18, line 360: The majority of our participants were high-functioning.

The authors are kindly asked to provide this information in material and methods under inclusion criteria. What means high-functioning precisely?

Page 18, lines 356-357: we chose to use both transducers simultaneously, rather than on separate occasions.

Indeed, this an important point. Pneumatic leg press, Gym Aware and Tendo Unit, SPPB, and Timed up and go (TUG) were separately explained on pages 6-9. The authors should provide detailed information of the sequence of assessments. For instance, were all assessments performed at the same occasions? Were all assessments arranged in the same order for all participants?

6. PLOS authors have the option to publish the peer review history of their article (what does this mean?). If published, this will include your full peer review and any attached files.

Reviewer #1: No

Reviewer #2: No

---

## [Author Response · Author response to Decision Letter 0]

17 Jul 2024

We have uploaded document with our specific responses to reviewers.

---

## [Editor Report · Decision Letter 1]

31 Jul 2024

Comparison of two popular transducers to measure sit-to-stand power in older adults

PONE-D-24-08546R1

Dear Dr. Balachandran,

We’re pleased to inform you that your manuscript has been judged scientifically suitable for publication and will be formally accepted for publication once it meets all outstanding technical requirements.

Kind regards,

Giuseppe Sergi

Academic Editor

PLOS ONE
---

## [Editor Report · Acceptance letter]

5 Aug 2024

PONE-D-24-08546R1 

PLOS ONE

Dear Dr. Balachandran, 

I'm pleased to inform you that your manuscript has been deemed suitable for publication in PLOS ONE. Congratulations! Your manuscript is now being handed over to our production team.

Kind regards, 

on behalf of

Dr. Giuseppe Sergi 

Academic Editor

PLOS ONE